# EFE-Mediated Ethylene Synthesis Is the Major Pathway in the Citrus Postharvest Pathogen *Penicillium digitatum* during Fruit Infection

**DOI:** 10.3390/jof6030175

**Published:** 2020-09-17

**Authors:** Ana-Rosa Ballester, Luis González-Candelas

**Affiliations:** Instituto de Agroquímica y Tecnología de Alimentos (IATA-CSIC), Calle Catedrático Agustín Escardino 7, 46980 Paterna, Spain; ballesterar@iata.csic.es

**Keywords:** postharvest pathology, citrus fruit, knockout mutant

## Abstract

*Penicillium digitatum* is the main fungal postharvest pathogen of citrus fruit under Mediterranean climate conditions. The role of ethylene in the *P. digitatum*–citrus fruit interaction is unclear and controversial. We analyzed the involvement of the 2-oxoglutarate-dependent ethylene-forming enzyme (EFE)-encoding gene (*efeA*) of *P. digitatum* on the pathogenicity of the fungus. The expression of *P. digitatum*
*efeA* parallels ethylene production during growth on PDA medium, with maximum levels reached during sporulation. We generated Δ*efeA* knockout mutants in *P. digitatum* strain Pd1. These mutants showed no significant defect on mycelial growth or sporulation compared to the parental strain. However, the knockout mutants did not produce ethylene in vitro. Citrus pathogenicity assays showed no differences in virulence between the parental and Δ*efeA* knockout mutant strains, despite a lack of ethylene production by the knockout mutant throughout the infection process. This result suggests that ethylene plays no role in *P. digitatum* pathogenicity. Our results clearly show that EFE-mediated ethylene synthesis is the major ethylene synthesis pathway in the citrus postharvest pathogen *P. digitatum* during both in vitro growth on PDA medium and the infection process, and that this hormone is not necessary for establishing *P. digitatum* infection in citrus fruit. However, our results also indicate that ethylene produced by *P. digitatum* during sporulation on the fruit surface may influence the development of secondary fungal infections.

## 1. Introduction

Ethylene (C_2_H_4_) is a gaseous compound involved in the regulation of numerous biological processes in plants, such as senescence, germination, flowering, and fruit ripening [1]. This hormone can be synthesized following three different pathways [2,3,4,5,6]: (i) in plants, methionine is transformed into S-adenosyl-L-methionine (SAM) by SAM synthetase. The subsequent reaction converts SAM into 1-aminocyclopane-1-carboxylic acid (ACC) by ACC synthase (ACS), followed by the conversion of ACC into ethylene by ACC oxidase (ACO). This three-step reaction is commonly known as the ACC pathway. (ii) In most bacteria and fungi that produce ethylene, the major pathway for its synthesis involves the oxidation of 2-keto-4-methylthiobutyric acid (KMBA), a transaminated derivate of methionine produced in a NADH:Fe(III)EDTA oxidoreductase-mediated two-step reaction (denoted as the KMBA pathway) [7]. KMBA may be converted into ethylene either enzymatically by peroxidase or spontaneously in the presence of light. The emission of ethylene via the KMBA pathway is associated with hyphal growth rather than the germination of *Botrytis cinerea* conidia [8]. KMBA can also be used as an ethylene precursor by the fungus *Penicillium digitatum* [7,9], although only trace amounts of ethylene are produced through this pathway. (iii) Ethylene can also be synthesized from 2-oxoglutarate and arginine by a single multifunctional ethylene-forming enzyme (EFE) in what is known as the 2-oxoglutarate pathway, which seems to be the predominant pathway in plant pathogenic *Pseudomonas* bacteria. 2-oxoglutarate is a common substrate produced by many organisms through the tricarboxylic acid cycle. This pathway produces the highest ethylene rates, and has been described in *Pseudomonas syringae* and *P. digitatum*, among other microorganisms [3,4,10,11]. Recently, a fourth ethylene synthesis pathway has been reported in phototrophic bacteria, in which ethylene may form in anoxic environments by an anaerobic methionine salvage pathway (MSP) that couples 5′-methylthioadenosine (MTA) metabolism to ethylene formation [12]. Besides its relevance as a plant hormone, it is worth mentioning that ethylene is one of the most produced organic compounds in the world and is used in the synthesis of different products, ranging from plastics, such as polyethylene and polyvinyl chloride (PVC), to textiles like polyester [11]. 

Ethylene production by fungi was first reported in 1940 from *P. digitatum* [13,14]. After that, ethylene production has been measured in many fungal species, such as *Aspergillus* sp., *Botrytis* sp., *Fusarium* sp., or *Verticillium* sp. [15]. The role of ethylene in fungus–host interactions is variable, depending on the interaction. Thus, ethylene may play an active role in plant resistance to invading pathogens. However, ethylene seems to facilitate fungal invasion in other instances [16,17]. Diverse studies have shown that fungi can contribute to ethylene production during the infection process, which occurs in the *B. cinerea*–tomato [8] or *P. digitatum*–citrus fruit [18,19] pathosystems. What has not yet been resolved is the question of whether the ethylene produced in planta by the fungus is required as a virulence factor [20,21,22,23]. For example, ethylene production by the necrotrophic fungal pathogen *Cochliobolus miyabeanus* increases rice susceptibility, which suggests a virulent role of fungal ethylene [20]. By studying a non-ethylene-producing *P. digitatum* isolate, Chalutz [24] concluded that ethylene, which is normally produced at very high rates by *P. digitatum* in vitro and in vivo, plays no clear role in the pathogenicity of this fungus because both the ethylene-producing isolate and the ethylene-deficient isolate infected citrus fruit. However, formal proof would require obtaining isogenic lines deficient in ethylene production. This could be achieved by knocking out a gene that participates in the pathway leading to ethylene synthesis. In *P. digitatum*, the KMBA and EFE pathways seem to co-exist [7,25,26]. Depending on the growing conditions, e.g., static or shaking cultures, ethylene synthesis is directed through one pathway or the other [27]. Regarding the genes involved in ethylene synthesis, the availability of the genome sequence of *P. digitatum* [28] has allowed for the identification of a putative gene coding for an EFE protein by Johansson et al. [29]. These authors demonstrated that the heterologous expression of this gene in *Saccharomyces cerevisiae* resulted in ethylene production, and so confirmed that this gene codes for a bona fide EFE enzyme. To the best of our knowledge, no single *P. digitatum* gene has yet been associated with ethylene synthesis through the KMBA pathway.

The present study aimed to gain further knowledge on ethylene production by *P. digitatum* during citrus fruit infection through the generation of a non-ethylene-producing *P. digitatum* knockout mutant.

## 2. Materials and Methods

### 2.1. Fungal Strains and Culture Conditions

*Penicillium digitatum* (Pers.: Fr.) Sacc. strain Pd1 (PDIP, deposited in the Spanish Type Culture Collection with accession code CECT20795) [28] was used as the parental strain to construct knockout mutants for the *efeA* gene. All the strains were grown on potato dextrose broth (PDB, Difco-BD Diagnostics, Sparks, MD, USA) or potato dextrose agar (PDA, Difco-BD Diagnostics, Sparks, MD, USA) plates, with or without the correspondent antibiotic. Cultures were incubated at 24 °C for 7–14 days. Conidia were scraped off the agar with a sterile spatula, suspended in sterile distilled water and titrated with a hemocytometer. 

### 2.2. Gene Expression Analysis

For the determination of *efeA* gene expression during *P. digitatum* in vitro growth, PDA plates covered with a cellophane membrane were inoculated centrally with 5 µL of a conidia suspension (10^5^ conidia/mL). Cultures were incubated at 24 °C for up to 14 days. Mycelium and spores were collected by scraping the surface of plates with a sterile spatula on days 1, 2, 3, 4, 5, 6, 7, 8, 11, and 14. Three biological replicates were collected per analysis day. Biomass was frozen in liquid nitrogen and kept at −80 °C until RNA extraction.

Total RNA was isolated from frozen spores and mycelia as previously described [30]. DNase treatment and first-strand cDNA synthesis were conducted using the Maxima H Minus cDNA synthesis kit with dsDNase (Thermo Scientific, Walthman, MA, USA) according to the manufacturer’s instructions. The expression analysis of *efeA* was performed by RT-qPCR (reverse transcription quantitative real-time PCR) as previously described [31], and following Minimum Information for Publication of Quantitative Real-Time PCR Experiments (MIQE) guidelines [32]. Gene-specific primer sets were designed for the gene expression analysis with Primer3Plus (Appendix A). For expression normalization, the β-tubulin gene (PDIP_27420) was employed as the reference gene. Relative gene expression (RGE) was calculated using the formula described by Pfaffl [33]. For each primer pair and each sample, PCR efficiency and the quantification cycle (Cq) were assessed by LinRegPCR software. RGE data were derived from three biological replicates, with two technical replicates each. 

For the determination of *efeA* gene expression during the infection process, sequence reads of *P. digitatum* spores and *P. digitatum*-infected oranges at 12, 24, and 48 h post inoculation (hpi) obtained using the Roche 454 GS FLX Titanium system [28] were retrieved from the Sequence Read Archive of the National Center for Biotechnology (NCBI, SRA ID: SRA059533). RNA-seq analysis was conducted using the RNA-seq module of CLC Genomics Workbench software v5.5.1 (CLC bio, Aarhus, Denmark). Trimmed reads were mapped to the *P. digitatum* Pd1 (PDIP) genome using default parameters, i.e., the number of mismatches allowed was two, minimum length fraction 0.5, and minimum similarity fraction 0.8. Gene expression values were calculated as reads per kilobase of exon model per million mapped reads (RPKM). 

### 2.3. Sequence and Phylogenetic Analysis

The sequence of the *efeA* gene was obtained based on the article by Johansson et al. [29]. These authors suggested revising the *P. digitatum* EFE protein sequence by modifying the start codon 147 base pairs upstream of the originally predicted start codon [28]. Sequences from the homolog fungal *efe* genes were obtained from the NCBI. A preliminary multiple protein sequence analysis for eight fungal selected sequences was performed with the MUSCLE algorithm [34]. Pfam domains were identified with CLC Genomics Workbench, version 12.0.3, using the Pfam-A database v33.1. A more extensive multiple protein sequence alignment with the MUSCLE algorithm [34] and the subsequent phylogenetic analysis were carried out by Molecular Evolutionary Genetics Analysis software (MEGA X version 10.1.8) [35]. The phylogenetic tree was inferred by using the maximum likelihood method and the Jones-Taylor-Thornton (JTT) matrix-based model. The bootstrap consensus tree was inferred from 1000 replicates.

### 2.4. Construction and Verification of the P. digitatum ΔefeA Knockout Mutants

Amplification of the *P. digitatum efeA* gene (PDIP_08660) was performed by PCR from the genomic DNA of the two gene flanking sequences using primers O1-O2 to amplify the upstream flanking region (amplicon of 1389 bp), and A3-A4 to amplify the downstream flanking region (amplicon of 1547 bp) (Appendix A). The resulting amplicons were cloned into pRF-HU2 by the uracil-specific excision reagent (USER) technique, as previously described [36]. The resulting plasmid, denoted as pRFHU-PdefeA, was introduced into *Escherichia coli* DH5α chemical-competent cells. Kanamycin-resistant transformants were screened by PCR for the presence of the promoter and terminator DNA fragments with primers pairs RF1-RF6 and RF2-RF5, respectively. Proper fusions were further confirmed by DNA sequencing. Then the plasmid was transferred to *Agrobacterium tumefaciens* AGL1 electrocompetent cells. Transformation of *P. digitatum* Pd1 was done as previously described [28], and transformants were selected on PDA supplemented with 100 µg/mL of hygromycin and 200 µg/mL of cefotaxime to avoid bacterial growth. The homologous recombination was verified by PCR using primers 1F-HPHTer2 (amplicon of 2104 bp) and 2R-HPHPro4 (amplicon of 1911 bp) for the promoter region and the terminator region, respectively (Appendix A). Further verification of the deletion of the target gene and the insertion of the hygromycin resistance marker was done with primers 3F-4R (amplicon of 471 bp) and HMBF1-HMBR1 (amplicon of 800 bp), respectively. The T-DNA copy number in the mutant strains was verified by quantitative real-time PCR (qPCR) (Appendix A). Briefly, analyses were performed with primers 5F-6R (amplicon of 108 bp), which are located in the region downstream of the gene between primer pairs A3 and A4, and normalized with a known single-copy gene coding for β-tubulin employing primer pairs betatubPDIG1 + betatubPDIG, as previously described [37]. All the samples were run in duplicate in a LightCycler 480 system (Roche Diagnostics, Basel, Switzerland) with SYBR Green to monitor DNA amplification.

### 2.5. Characterization of the ΔefeA Knockout Mutants: Mycelial Growth, Sporulation, and In Vitro Ethylene Production

To quantify the mycelial growth and sporulation of *P. digitatum* Pd1, the ectopic *efeA* mutant, and two Δ*efeA* knockout mutants, 5 µL of the conidial suspensions adjusted to 10^6^ conidia/mL were centrally inoculated onto PDA plates and incubated in the dark at 24 °C. Mycelial growth was determined by measuring two perpendicular diameters of growing colonies up to 6 dpi. The sporulation assessment was done as described in Section 2.1.

Ethylene production was determined in three replicate samples of PDA plates centrally inoculated with 5 µL of the conidial suspensions (10^6^ conidia/mL) of parental strain *P. digitatum* Pd1, the ectopic mutant and two knockout mutants. Inoculated plates were incubated at 24 °C for up to 14 days. To determine ethylene production, each PDA plate was incubated without the lid in a 1 L sealed glass jar at 24 °C for 1 h. Then 1 mL of headspace gas was drawn from each jar with a hypodermic syringe and injected into a 7820A gas chromatograph system 7820A (Agilent Technologies, Santa Clara, CA, USA), equipped with a HayeSep Q pre-conditioned column (0.91 m long, 2 mm inner diameter, mesh size of 80/100) and a flame ionization detector [38]. Nitrogen was used as the carrier gas, and the column temperature was maintained at 140 °C. The ethylene standard was obtained from Abello-Oxigeno-Linde, S.A. (Valencia, Spain). The results are indicated as the mean of three replicate samples ± the standard error of the mean (SEM).

### 2.6. Ethylene Production during P. digitatum Growth on Orange Discs

To determine ethylene production by *P. digitatum* when using orange peel tissue as a nutrient source, oranges (*Citrus sinensis* cv ‘Navel’) were obtained from a packinghouse in Liria, Valencia (Spain) on the same harvesting day before receiving any postharvest treatment. Fruit were washed for 5 min in a 5% commercial bleach solution, thoroughly rinsed with tap water, and allowed to dry until the next day. Orange peel discs were obtained with a 5 mm cork-borer and subjected to three flash-freezing cycles in liquid nitrogen and thawing at ambient temperature. This procedure inactivated orange cells, as noted by a lack of respiration. Five discs were placed inside 15 mL glass tubes. Then orange discs were inoculated with 500 µL of a conidia suspension (10^6^ conidia/mL) of parental strain Pd1 or the Δ1f knockout mutant. Five replicate tubes were incubated at 24 °C for up to 21 days. Ethylene production was determined by sealing the tubes containing the inoculated discs and incubating them at 24 °C. After 1 h of incubation, a 1 mL headspace gas sample was withdrawn from each tube, and ethylene production was determined as previously described.

### 2.7. Citrus Fruit Pathogenicity Assays

To analyze the virulence of the generated Δ*efeA* knockout mutants, pathogenicity assays were conducted with citrus fruit. *C. sinensis* ‘Navelina’ oranges were harvested at the beginning of the season and used for experiments 1 and 2. The ‘Lane-Late’ oranges employed for experiment 3 were harvested at the end of the season. Oranges were obtained from a packinghouse in Liria, Valencia (Spain) on the same harvest day and taken to the laboratory, where they were washed as described before. The next day, four equidistant wounds (3 mm deep) were made along the equatorial axis and were immediately inoculated with 10 µl of a conidial suspension, adjusted to 10^4^ conidia/mL, from the parental *P. digitatum* or the ectopic or the Δ*efeA* knockout mutant strains. Five fruit constituted a replicate, with three replicates per treatment. Fruit were incubated in a thermostatic chamber at 20 °C and 90% relative humidity (RH) for up to 7 days. The incidence, measured as the percentage of infection, and infection severity, measured as decayed area (cm^2^) for all inoculated wounds, were determined up to 7 dpi. An analysis of variance (ANOVA) was performed to test the differences among samples. Means were separated using Tukey’s test with *p* < 0.05 by employing Statgraphics Stratus (Statgraphics Technologies, Inc., The Plains, VA, USA).

### 2.8. Ethylene Production during Orange Infection by P. digitatum

To determine the possible involvement of ethylene biosynthesis in the pathogenicity of *P. digitatum*, fully mature oranges (*C. sinensis* cv ‘Navel’) were obtained and processed as described before. In this experiment, the inoculum was raised to 10^6^ conidia/mL to obtain a more synchronous infection. Unwounded and mock-inoculated samples were also used as controls. Fruit were kept at 20 °C and 90% RH. Ethylene production from whole fruit was measured periodically by incubating three biological replicate samples containing five fruit with four inoculated wounds each in 3.8 L sealed glass jars for 1 h at 24 °C. Two technical replicate samples of the 1 mL headspace gas sample were withdrawn from each glass jar, and ethylene production was determined as previously described.

## 3. Results

### 3.1. Identification and Phylogenetic Analysis of Fungal efe Genes

The 2-oxoglutarate-dependent ethylene-forming enzyme (EFE) involved in ethylene biosynthesis has been previously identified in *P. digitatum* [29]. These authors realized that the start codon of the *efeA* gene should be located 147 base pairs upstream of the predicted start codon in PDIP_08660 in the published annotated genome of *P. digitatum* [28]. The MUSCLE alignment of the previously confirmed ethylene-forming enzymes from *P. syringae* (P32021.1) and *P. digitatum* (PDIP_08660), as well as similar genes from *Penicillium expansum* (XP_016595579.1, PEX2_047820), *Penicillium griseofulvum* (KXG45570.1, PGRI_033370), *Penicillium rubens* (XP_002562422.1), *Penicillium camemberti* (CRL20948.1, PCAMFM013_S005g000112), *Aspergillus clavatus* (XP_001270369.1, ACLA_098850), *Fusarium oxysporum* (PCD42226.1), and the scanning of these protein sequences against the Pfam database showed they all contained the non-haem dioxygenase in morphine synthesis N-terminal (DIOX_N, PF14226) and the 2OG-Fe (II) oxygenase superfamily (2OG-FeII_Oxy, PF03171) domains, located between amino acids 63-181 and 202-345 in the *P. digitatum* EFE protein, respectively (Figure 1A). The sequences from the homolog *efe* fungal genes were retrieved from the NCBI. Phylogenetic tree reconstruction based on protein sequences from the significant BLAST hits flanked the *Penicillium* proteins with the *Aspergillus* and *Colletrotrichum* ones by grouping the *Fusarium* and *Verticillum* proteins in another clade (Figure 1B).

### 3.2. Characterization of P. digitatum efeA Gene Expression

We analyzed the expression of the *efeA* gene during the in vitro *P. digitatum* growth (Figure 2A). PDA plates were centrally inoculated with a spore dilution (10^5^ conidia/mL) of *P. digitatum* and incubated at 24 °C for up to 14 days. Mycelia and spores were scraped from the surface of plates, and RNA was extracted and cDNA synthesized. The *efeA* gene expression was low in spores and increased during radial *P. digitatum* growth on PDA plates by up to 25- and 39-fold, with maximal values at 8 and 11 days, respectively. Later, the relative *efeA* gene expression decreased by 14 days.

We determined the *efeA* gene expression in *P. digitatum* spores and *P. digitatum*-infected oranges at 12, 24, and 48 hpi using available RNA-seq data (NCBI, SRA ID: SRA059533) [28]. Figure 2B shows the reads per kilobase of exon model per million mapped reads (RPKM), corresponding to the *efeA* gene in spores (S), and at 12, 24, and 48 h post inoculation (hpi). The *efeA* gene expression was detected only in spores.

### 3.3. Phenotypic Characterization of the Mutants

*A. tumefaciens*-mediated transformation of *P. digitatum* was conducted to obtain *efeA* deletants. Two Δ*efeA* gene knockouts, named Δ1f and Δ23c, and an ectopic mutant were selected after the analysis of putative transformants (Appendix A). No deletant mutant contained any extra T-DNA copies (Appendix A). To phenotypically characterize the knockout mutants, a spore dilution (10^6^ conidia/mL) of parental strain Pd1, the ectopic mutant (ect), and two knockout mutants (Δ1f and Δ23c) was point inoculated in the center of PDA plates and incubated in the dark at 24 °C. No differences were observed in mycelial growth among the different strains at 6 dpi (Table 1). Similar amounts of conidia were measured at 7 and 14 dpi among the different strains (Figure 3A).

To address whether the ethylene-forming enzyme-encoding *efeA* gene is involved in ethylene biosynthesis by *P. digitatum*, ethylene production was determined daily while the four strains grew on the PDA plates incubated at 24 °C (Figure 3B). Ethylene production increased during incubation, with the highest production at 7 and 8 dpi for the ectopic mutant and the Pd1 parental strain, respectively, when PDA plates were completely covered with conidia. A sudden drop in ethylene production was observed later on up to day 14, which was the last day of the experiment. However, ethylene production did not increase up to 14 dpi in either knockout mutant.

### 3.4. P. digitatum ΔefeA Knockout Mutant does not Produce Ethylene during Growth on Orange Peel Discs

It is known that citrus fruit produce ethylene in response to different abiotic and biotic stresses, such as wounding and *P. digitatum* infection [38,39,40]. To determine the ethylene produced by the fungus without the ethylene produced by fruit during the infection process interfering, five inactivated 5 mm orange peel discs were inserted into a 15 mL glass tube, inoculated with a spore dilution (10^6^ conidia/mL) of the parental Pd1 strain and the Δ1f knockout mutant, and incubated for up to 21 days at 24 °C. Development of parental strain Pd1 on orange discs led to a sharp increase in the ethylene levels, starting at 3–4 dpi, with a maximum production of 280 nL/g h at 11 dpi (Figure 4A). Ethylene production decreased thereafter until 21 dpi. For the knockout mutant–orange disc interaction, no ethylene production was detected up to 21 dpi under our experimental conditions. The Δ1f knockout mutant failed to produce ethylene either on PDA (Figure 3B) or orange peel discs (Figure 4A) throughout mycelium and spore development. The mycelial growth of both parental strain Pd1 and the Δ1f knockout mutant was visible at 3 dpi when the rise in ethylene production started for the fungus–orange disc interaction (Figure 4B). Sporulation was observed for both strains at 5 dpi and increased during pathogen development.

### 3.5. Ethylene Production by P. digitatum Is Dispensable for Orange Fruit Infection

Previous reports have shown that ethylene does not play a role in the pathogenicity of *P. digitatum* when citrus fruit were inoculated with either a producing or a natural non-ethylene producing isolate [18,24]. To evaluate the role of ethylene in virulence, two independent pathogenicity tests were run using orange fruit with parental strain Pd1, the ectopic mutant, the non-ethylene producing Δ1f, and Δ23c knockout mutants. Regarding incidence, our results showed that the knockout mutants were as virulent as parental strain Pd1 and the ectopic mutant (Figure 5A). However, the disease severity, based on macerated areas around infected wounds, of the Δ23c knockout mutant was lower than that observed for the parental strain, the ectopic mutant, or the Δ1f knockout mutant in two out of the three experiments (Figure 5B).

### 3.6. Changes in Ethylene Production during the Orange Fruit Infection by P. digitatum

To determine changes in ethylene production during infection progress, oranges were inoculated with either parental strain Pd1 or the Δ1f knockout mutant (Figure 6A). The unwounded and mock-inoculated oranges were used as controls. When oranges were inoculated with parental strain Pd1 and the Δ1f knockout mutant, similar amounts of ethylene were detected in early infection stages, between days 0 to 6, with a maximum at 3 dpi followed by a decrease. This ethylene production was not observed in the unwounded or the mock-inoculated oranges. Afterward, a marked increment in ethylene production was observed in the fruit inoculated with ethylene-producing parental strain Pd1, with 500-fold higher ethylene production by 14 dpi compared to the peak detected by 3 dpi (Figure 6A). However, ethylene did not increase in the fruit inoculated with the knockout mutant, the mock-inoculated fruit, or unwounded fruit. The rise in ethylene production observed in the fruit inoculated with the parental strain coincided with the development of mycelia and spores on fruit surfaces starting at 5 dpi (Figure 6B). The ethylene produced by the Pd1-infected oranges decreased at the end of the experiment, at 21 dpi. At this time point, the oranges inoculated with the non-ethylene producing knockout mutant were completely covered by other bluish spore fungi, which were present to a much lesser extent in the Pd1-infected oranges (Figure 6C,D).

## 4. Discussion

Ethylene is a hormone that plays a controversial role in fungus–plant interactions [17]. In some instances, its production is needed to fine-tune the plant responses against the attacker, whereas it seems to favor the development of the pathogen in other pathosystems, especially in the case of necrotrophs. This hormone is released during plant infection by pathogenic fungi, and there is evidence showing that both the plant and fungus may contribute to ethylene production [18,24]. This hormone can act as a plant defense factor or might have a direct effect on the pathogen [16]. For example, ethylene enhances the spore germination of *P. digitatum* under some culture conditions [41], or the germination and appressorium formation of *Colletotrichum* spp., which are pathogens of climacteric fruit [42]. In the present study, we constructed a *P. digitatum* non-ethylene producing mutant by knocking out the *efeA* gene involved in ethylene biosynthesis. We used the non-ethylene producing mutant to determine the prevalent ethylene biosynthetic pathway in this necrotrophic fungus and to study the role of this hormone in the *P. digitatum*–orange fruit interaction.

### 4.1. The 2-Oxoglutarate EFE-Mediated Pathway Is the Major Ethylene Biosynthetic Pathway Used by P. digitatum during in Vitro Growth on PDA

Three different ethylene biosynthetic pathways have been described in nature: (i) the ACC pathway, mainly used by higher plants and uncommon in microorganisms; (ii) the KMBA pathway, employed by bacteria and fungi; (iii) the 2-oxoglutarate pathway, described in *P. syringae* and *P. digitatum*, among others. Some authors have attributed the capability of producing ethylene to *P. digitatum* using any of these three pathways [15] based on the fact that a closely related species, *Penicillium citrinum*, was found to synthesize ACC [43] through the action of an ACC synthase [44]. Besides the fact that *P. digitatum* and *P. citrinum* are two different species [45], *P. citrinum* does not produce ethylene; it degrades ACC into 2-oxobutyrate and ammonia by the action of an ACC deaminase [46]. Moreover, it has been demonstrated that the addition of ACC leads to a marked increase in ethylene production by healthy citrus peel tissue, but it does not affect ethylene production by *P. digitatum*-infected oranges [18]. The addition of ACC or the ethylene inducer indoleacetic acid or the ethylene inhibitor aminooxyacetic acid to the media culture does not change the ethylene produced by *P. digitatum* [26]. These results reinforce the idea that *P. digitatum* is not able to synthesize ethylene by the ACC pathway.

There are reports that ethylene can be formed by *P. digitatum* by using the KMBA and EFE pathways, depending on growth conditions [26]. This fungus produces KMBA in culture fluids and, in the presence of methionine, glucose enhances ethylene production in shake cultures [7,9]. Regarding the 2-oxoglutarate pathway, Johansson et al. [29] identified EFE in *P. digitatum* and showed its capability to mediate ethylene production in yeast. Our data reveal that the expression of the *efeA* gene of parental strain *P. digitatum* Pd1 increased concomitantly with pathogen development on PDA plates. Moreover, the parental strain produced ethylene during in vitro growth on PDA, and gradually increased in parallel to fungal growth and sporulation. However, the non-ethylene producing Δ*efeA* knockout mutant did not produce ethylene above the background levels in any fungal development stage. Previous studies by Yang et al. [26] have suggested that *P. digitatum* uses the 2-oxoglutarate pathway for ethylene synthesis when grown in PDA tubes with low oxygen levels and limited nutrient resources. However, these authors also suggested that when the fungus was grown on PDA plates, the KMBA pathway was mainly responsible for ethylene synthesis. This conclusion was built upon the induction effect exerted by methionine, a precursor in the KMBA pathway. Our results clearly show that under the assayed conditions, and with the PDA medium incubated in the dark and no potential ethylene precursor added, most, if not all, the ethylene produced by *P. digitatum* was via the 2-oxoglutarate/EFE pathway. However, we cannot rule out the possibility of the KMBA pathway making a minor contribution to ethylene synthesis under our experimental conditions.

### 4.2. Ethylene Is Synthesized by EFE in P. digitatum during Citrus Infection, but Is Dispensable for Establishing Infection

Ethylene may play a dual role in fungus–plant interactions by affecting both the plant and the pathogen. This hormone has been implicated in biotic stresses as both a virulence factor of fungal pathogens and a signaling compound in disease resistance [16,17,47]. The necrotrophic fungus *Alternaria alternata* is able to produce ethylene via the KBMA pathway and to utilize it for enhanced virulence during grape berry infection [48]. *B. cinerea* also produces ethylene via the KMBA pathway, and the emission of the hormone follows the hyphal growth pattern [2,8]. However, ethylene levels were higher during tomato infection compared to in vitro growth. Besides, exogenous ethylene promotes hyphal growth and pathogenesis in the *B. cinerea*–grape system [49]. *Verticillium dahliae* produces ethylene during in vitro growth and the role of the hormone during the infection process depends on timing: ethylene presence upon infection inhibits disease development, whereas it enhances symptoms in tomato [50]. In the *P. digitatum*–citrus fruit interaction, the ethylene detected in initial infection stages has been described to be of fruit origin, while the ethylene burst observed in advanced development stages originates from the fungus [18,19]. Chalutz [24] identified a natural non-ethylene-producing isolate of *P. digitatum* and used it to test the role of ethylene in the pathogenicity on citrus fruit. He showed that the natural isolate did not produce ethylene during in vitro growth and was as pathogenic as the ethylene-producing isolate. However, as the mutations present in the non-ethylene producing isolate were not known, this result does not constitute formal proof to rule out any role of ethylene in *P. digitatum* pathogenesis. During the infection of *Citrus reticulata* fruit by *P. digitatum*, at least five fungal genes associated with ethylene synthesis were up-regulated [51]. As far as we know, of these, only PDIP_08660, which encodes EFE, has been identified in *P. digitatum* as a gene involved in ethylene biosynthesis by using 2-oxoglutarate as a precursor [29]. In order to clarify the role of ethylene in the pathogenesis of *P. digitatum*, we constructed a knockout mutant of the *efeA* gene. The Δ1f knockout mutant and parental strain Pd1 are isogenic strains with similar morphological characteristics and growth rates both in vitro and in vivo. Moreover, the knockout mutant was as pathogenic as the parental high ethylene-producing isolate during citrus fruit infection, which confirms previous results obtained with a natural non-ethylene-producing isolate [24]. According to previous findings [18,19,24], in our experiments, fruit inoculation either by the parental strain or the non-ethylene-producing Δ*efeA* knockout mutant resulted in a slight increase in ethylene production in the initial infection phases. This enhanced production of ethylene originates mainly from the citrus fruit as a response to the pathogen. Marcos et al. [39] showed that the burst in ethylene production in citrus fruit upon *P. digitatum* infection resulted from the coordinated distinct spatio-temporal up-regulation of the three ethylene biosynthetic genes *ACS1*, *ACS2*, and *ACO* in citrus fruit. This initial rise was followed by a more pronounced increase in the fruit infected with the *P. digitatum* parental strain. The increased ethylene production in advanced infection states is likely to stem mostly from the fungus because this ethylene burst was observed only in the fruit infected by parental strain Pd1. The ethylene burst was not detected in the fruit infected by the non-ethylene-producing knockout mutant, which reinforces the idea that the ethylene generated in advanced infection stages is produced mainly by the fungus and not by citrus fruit. Moreover, as the knockout mutant only lacks the *efeA* gene, we can affirm that the ethylene produced by parental strain Pd1 in advanced infection stages of citrus fruit is synthesized through the 2-oxoglutarate/EFE pathway. Although it would seem clear that ethylene synthesis by *P. digitatum* is dispensable for establishing the colonization of citrus fruit, the role of ethylene in advanced development stages of infection is still unclear. Our results suggest that the inoculation of oranges with an ethylene-deficient mutant facilitates the development of secondary pathogens on fruit surfaces in advanced infection stages, when fruit surfaces are completely colonized by spores. This secondary colonization was present to a much lesser extent when oranges were infected with the ethylene-producing parental strain. We hypothesize that the ethylene produced by the parental strain could serve to deter the development of opportunistic pathogens that could compete with *P. digitatum* for the nutrients released by the killed fruit tissue. Further experiments should be performed to explore this possible new role of ethylene produced by *P. digitatum* during orange fruit infection.

## 5. Conclusions

The construction of a *P. digitatum* Δ*efeA* knockout mutant allowed us to analyze the role of the 2-oxoglutarate pathway in ethylene synthesis in this citrus postharvest pathogen. A lack of ethylene production observed in the mutant when grown on PDA medium in the dark led us to conclude that this pathway is responsible for most, if not all, the produced ethylene. During citrus fruit infection with the wild-type strain, a marked rise in ethylene production was observed at the time of mycelia and spore colonization on the fruit surfaces. However, this ethylene burst was not observed when infection was carried out by the Δ*efeA* knockout mutant. This result indicates that the 2-oxoglutarate pathway also constitutes the main route for ethylene synthesis in this fungus during the infection process. Finally, we observed a widespread presence of bluish spores on the surfaces of the fruit colonized by the Δ*efeA* knockout mutant, which is indicative of the development of secondary infections. These secondary infections were less abundant when fruit were colonized by the parental strain. This result suggests a new possible role for ethylene in the infection process as a tool to deter the development of secondary pathogens on the surface of already dead fruit tissue, and precludes other organisms from taking advantage of the presence of the readily available source of nutrients. Nevertheless, further work is necessary to explore the feasibility of this hypothesis.

## Figures and Tables

**Figure 1 jof-06-00175-f001:**
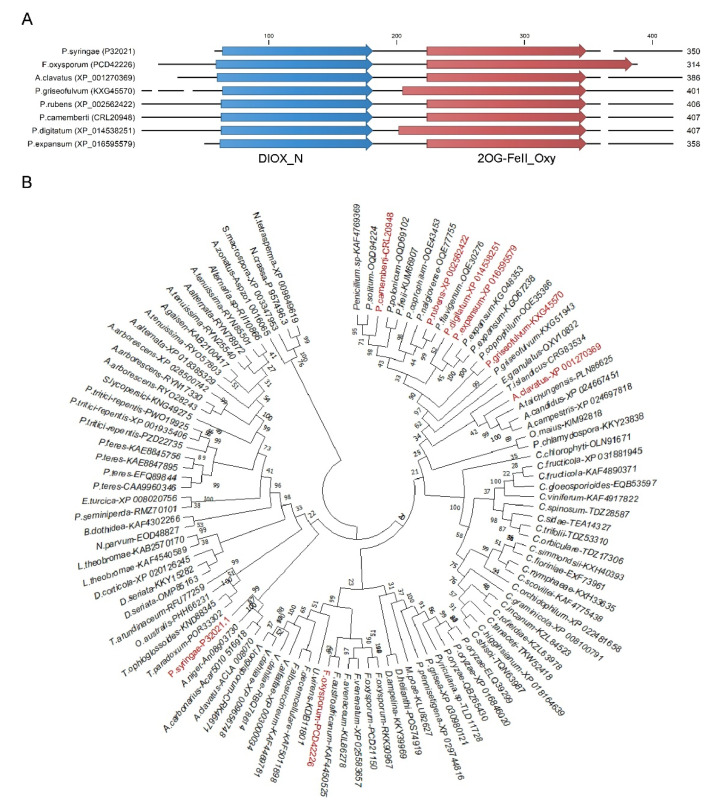
Protein domains and phylogenetic tree of the ethylene-forming enzymes (EFE) proteins from filamentous fungi. (**A**) A multiple protein sequence alignment of the seven selected fungal EFE proteins, plus that from bacterium *P. syringae*, was performed with MUSCLE. The non-haem dioxygenase in morphine synthesis N-terminal (PF14226.7) and the 2OG-Fe (II) oxygenase superfamily (PF03171.21) domains are represented as blue and red arrows, respectively. (**B**) Multiple protein sequence alignments were performed with the MUSCLE algorithm from MEGA X. The phylogeny tree of the EFE proteins of filamentous fungi was inferred by the maximum likelihood method and the Jones-Taylor-Thornton (JTT) matrix-based model. The bootstrap consensus tree inferred from 1000 replicates is taken to represent the evolutionary history of the analyzed taxa. This analysis involved 105 protein sequences, with a total of 1215 positions in the final dataset. Evolutionary analyses were conducted in MEGA X. The fungi used for the multiple sequence alignment in (**A**) are indicated in red. Appendix A includes further details of all the sequences employed in this analysis.

**Figure 2 jof-06-00175-f002:**
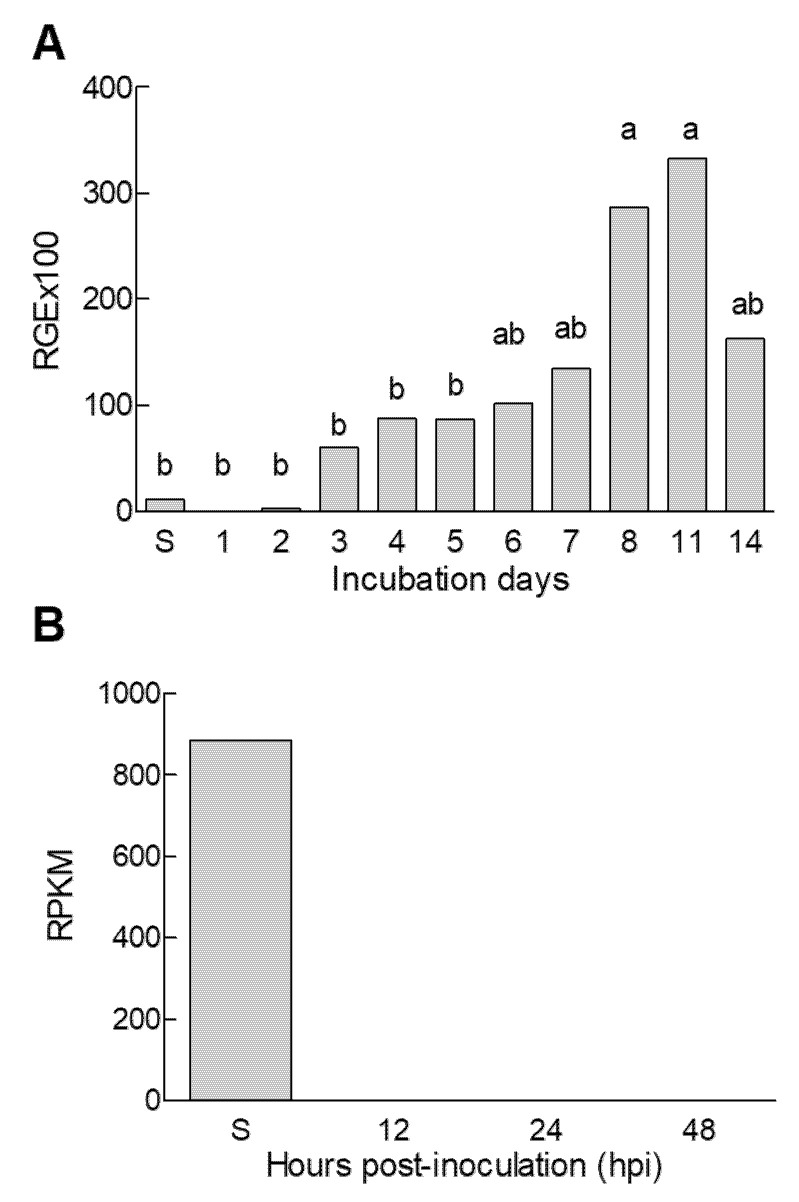
Gene expression analysis of the *P. digitatum efeA* gene. (**A**) Relative gene expression (RGE) of the *efeA* gene (PDIP_08660) during growth on potato dextrose agar (PDA) medium. The spore suspension (5 µL of 10^5^ conidia/mL) was inoculated on the center of PDA plates and incubated for up to 14 days at 24 °C in the dark. Expression levels are relative to the β-tubulin reference gene. Values are the mean of at least three biological replicates. Different letters indicate significant differences among samples according to Tukey’s test with a *p*-value of 0.05. (**B**) Reads per kilobase of exon model per million mapped reads (RPKM) values from RNA-seq during orange fruit infection by *P. digitatum* using the Roche 454 GS FLX Titanium system [28]. Samples analyzed: *P. digitatum* spores collected after 7 days of growth in PDA (S), and *P. digitatum*-infected oranges at 12, 24, and 48 h post inoculation (hpi).

**Figure 3 jof-06-00175-f003:**
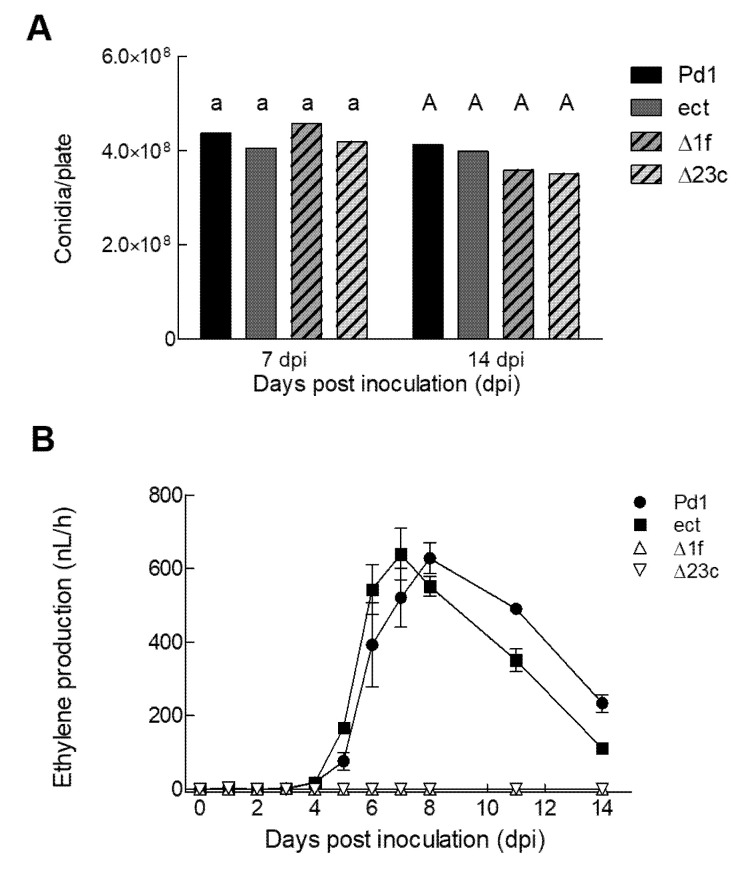
Phenotypic characterization of the *P. digitatum* Δ*efeA* knockout mutants. (**A**) Conidia production of the parental *P. digitatum* Pd1 strain, the ectopic mutant and two knockout mutants. Strains were point inoculated (10^6^ conidia/mL) in the center of potato dextrose agar (PDA) plates, and conidia were collected and counted after 7 and 14 incubation days in the dark at 24 °C. The values represent the mean of at least three biological replicates. The bars labeled with the same letter at each time point do not differ at the 95% confidence level based on Tukey’s honest significant difference (HSD) procedure. (**B**) Ethylene production by the different strains grown on PDA plates and incubated at 24 °C for up to 14 days post inoculation (dpi). Ethylene production is expressed as nL/h per plate. The error interval indicates the standard error of the estimated mean value. The ethylene production values are included in Appendix A.

**Figure 4 jof-06-00175-f004:**
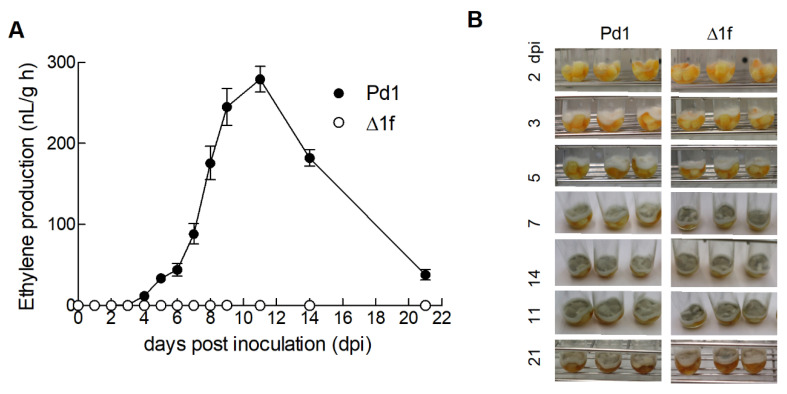
(**A**) Ethylene production during the growth of parental strain Pd1 (●) and the Δ1f knockout mutant (○) on inactivated orange peel discs. Orange peel discs were inoculated with a conidia suspension (10^6^ conidia/mL) of both strains in 15 mL glass tubes at 24 °C for up to 21 days post inoculation (dpi). Values are the mean of five biological replicates ± SEM. The ethylene production values are included in Appendix A. (**B**) Images show the development of parental strain Pd1 and the Δ1f knockout mutant on the orange peel discs incubated at 24 °C up to 21 dpi.

**Figure 5 jof-06-00175-f005:**
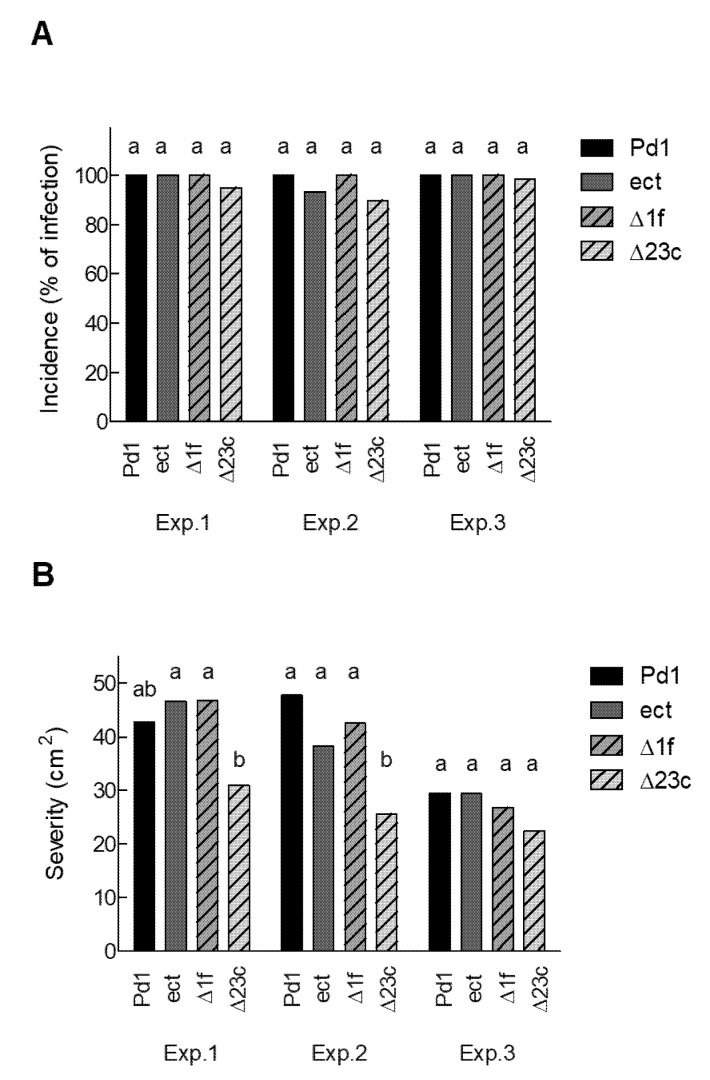
(**A**) Incidence (measured as the percentage of infected wounds) and (**B**) severity (measured as the macerated area in cm^2^) in the oranges infected with the parental *P. digitatum* Pd1 strain, the ectopic mutant, or the two knockout mutants in three independent experiments. Experiments 1 and 2 were carried out at the beginning of harvest, while experiment 3 was performed at the end of harvest. Oranges were inoculated with 10 µL of a spore suspension (10^4^ conidia/mL) of the different strains. Fruit were incubated at 20 °C and 90% relative humidity for up to 7 days. There were three replicates of five fruit and four wounds per fruit. Bars show the mean values of both incidence and severity 7 d post inoculation. The bars labeled with the same letter in each experiment do not differ at the 95% confidence level based on Tukey’s honest significant difference (HSD) procedure.

**Figure 6 jof-06-00175-f006:**
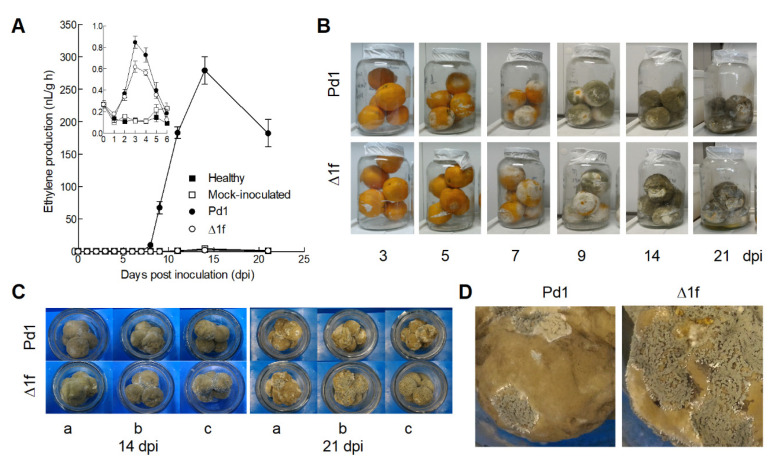
(**A**) Ethylene production by the *P. digitatum*-infected whole oranges with parental strain Pd1 (●) and the Δ1f knockout mutant (○). Control fruit were mock-inoculated (□) or left unwounded (■). Five fruit were placed inside a 3.8 L glass jar and kept at 24 °C. Jars were sealed daily for 1 h before sampling the air above the fruit for ethylene determination. The insert panel shows the values corresponding to the ethylene production of the parental strain, the knockout mutant, the control and the mock-inoculated samples on a smaller scale to better illustrate the differences found among these samples up to 6 dpi. The values represent the average of three biological replicates ± SEM. (**B**) The images show the development of parental strain Pd1 and the Δ1f knockout mutant on the oranges stored at 20 °C for up to 21 dpi. (**C**) A detailed view of the oranges infected with parental strain Pd1 and the Δ1f knockout mutant of the three biological replicates after incubation at 20 °C for 14 and 21 dpi. (**D**) A close-up image of the oranges infected with the parental strain and the knockout mutant at 21 dpi to facilitate the visualization of secondary infections, the bluish spore fungus.

**Table 1 jof-06-00175-t001:** Colony diameter (in mm) of the parental *P. digitatum* Pd1 strain, the ectopic mutant and the two knockout mutants.

Strain	2 dpi	3 dpi	4 dpi	6 dpi
Pd1	12.3 ^ab^	23.0 ^b^	31.4 ^ab^	53.0 ^a^
ect	11.4 ^b^	21.1 ^c^	30.9 ^b^	53.0 ^a^
Δ1f	11.6 ^b^	22.8 ^b^	32.1 ^ab^	53.0 ^a^
Δ23c	13.3 ^a^	24.3 ^a^	33.8 ^a^	53.0 ^a^

Strains were point inoculated (10^6^ conidia/mL) in the center of potato dextrose agar (PDA) plates, and two perpendicular colony diameters per plate were recorded 2, 3, 4, and 6 days post inoculation (dpi). Plates were incubated in the dark at 24 °C. The values represent the mean of at least three biological replicates, and are those labeled with the same letter at each time point do not differ at the 95% confidence level based on Tukey’s honest significant difference (HSD) procedure.

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
