# Peer review of "EFE-Mediated Ethylene Synthesis Is the Major Pathway in the Citrus Postharvest Pathogen *Penicillium digitatum* during Fruit Infection"

_jof, 2020, doi:10.3390/jof6030175_

Round 1

Reviewer 1 Report

This paper is well written. I think it is suitable for publication in Journal of Fungi.

Reviewer 2 Report

The manuscript titled "EFE-mediated ethylene synthesis is the major pathway in the citrus postharvest pathogen Penicillium digitatum during fruit infection" studied the role of the 2-oxoglutarate pathway in the synthesis of ethylene in this citrus postharvest pathogen. The construction of a P. digitatum efeA knockout mutant has allowed them to analyze the role of the 2-oxoglutarate pathway  and conclude that this pathway could be responsible for most if not all ethylene produced on PDA and on mature oranges. Although other authors (Yang et al. 2017) suggested that only the KMBA pathway seemed to be activated under other culture conditions (i.e. on PDA plates). In other hand, the use of mutant could be suggest a new possible role for ethylene in the infection
process as a tool to deter the development of secondary pathogens on the surface of the already dead fruit tissue, precluding other organisms to take advantage of the presence of the readily available source of nutrients.

The manuscript is correctly designed and the tools developed allow the authors to obtain interesting results on the already extensive bibliography on ethylene in necrotrophic fungi. I believe that the article has sufficient merit to be published in the Journal of Fungi, as long as the following questions are clarified in its revision:

1.- L83-85, the results should not be included in the work objectives

2.- Why did you suggest a revision of the protein sequence of EFE for P. digitatum that contains a mitochondrial signal peptide compared to the one previously annotated in the published genome? Please clarify

3.- Please clarify RNAseq results (Lines 246-247; 251-254). If you used these RNAseq data, maybe you must describe them in M&M section

4.- The mutants obtained 1f and 23c seem to behave the same in terms of growth, sporulation and incidence of the disease in fruit. However, oranges infected with 23c have less severity of the disease (Lines 359-369), could you explain this?

5.- The data in Figures 2 and 4 have been analyzed by means of an analysis of variance and a comparison of means by the Turkey test. The letters on the bars show these differences. It is not necessary to show the standard error on the bars, since when performing an ANOVA it is assumed that there is necessarily a homogeneity of the variance.

6.- The authors should explain the disagreements with other authors regarding the route of ethylene production by P. digitatum in PDA.

Reviewer 3 Report

In this work, the authors present a genetic study of the role of ethylene in p. digitatum virulence on orange. Most of the study is well designed and executed. I do think that to conclusively say that ethylene has no role in virulence, the authors need to investigate in more depth why they had decreased virulence in one of their knockout mutants but not the other.

Did the authors examine the functionality of the other ethylene pathway (the one not mutated) in the mutant lines (e.g., by gene expression?)? or are there no genes known to be involved in ethylene biosynthesis there?

Additionally, some of the data appear to be provided in the text without being presented in the manuscript itself in result format- descriptions of the growth pattern, conidia amount, or secondary infections observed- all without any quantification or imaging- are not acceptable. please provide analyses of fungal mass, conidia produced per particular fungal mass, and quantification of secondary infection. where experiments were conducted twice- please add a third repeat.

Comments:

Figure 2: line 243: what's the significance of "1100" level of gene expression? please explain the values used, and consider whether normalization would be beneficial.

Figure 2B: were the samples normalized for fungal amount? despite the use of a normalizing fungal gene, at very low levels of fungus qPCR results can be unreliable.

line 252: please avoid the use of data not shown. data can be included in a supplement or removed.

lines 268-295: depending on how routine (or not) transforming p. digitatum is, this entire section could be moved to the supplement.

Figure 4: please include the legend for 4a in the figure itself. lines 301-302: it appears that the reduction in conidia production in the deletion mutants is not statistically significant, so is not, per se, a reduction. please consider this point; were the etyhlene assays normalized for fungal mass?

The wt and ectopic strain produce very high ethylene levels, but do the deletion mutants produce zero or trace amounts that are measurable? please either include the data in table form (with statistics) in the supplement or adjust the Y-axis scale to be bi-partite so that the ethylene levels in the deletion mutants can be accurately presented to the reader. Or: please include a graph showing total ethylene production, or ethylene production on a particular date.

Figure 5: Did the authors check the fungal mass/ spore amount in the experiments with orange peels? indeed, was fungal mass measured and compared between the mutants and the parental line at all? this seems important.

Figure 6: again, please include the legend in the figure. Why are both experiments presented separately? Given the slightly different behavior of the ectopic line, it seems like another repeat would be in order, particularly since the two mutants behave differently.

Figure 7: why did the authors use only the 1f mutant, when the other mutant appears to have a decrease in virulence? Figure 7A is a perfect example of what I was saying about ethylene levels in connection with figures 4 and 5.

panel C in figure 7 is not clear at all, perhaps you could do some close-up imaging (e.g, stereomicroscope?)? perhaps there are differences at a smaller scale?

Abstract: is p. digitatum really the main post-harvest fungal pathogen in citrus? worldwide? perhaps change to "an important" pathogen or something similar.

Conclusions: this section is rather short, I am missing an in-depth discussion of the results. The brief description of secondary infections without any quantification or imaging is rather unfortunate. Frustratingly, it cannot allow the reader the benefit of sharing the conclusion with the authors.

English corrections (these are just examples, please do some grammar editing throughout):

  • line 336: replace "during the stages of " with "throughout"
  • line 338: replace "on" with "in"
  • line 510: "led" or "leads", not "lead"
  • line 521: replace "to take" with "from taking"

Round 2

Reviewer 3 Report

The authors have adequately answered most of my queries.

Two questions:

(1) Were the details of the RNAseq added to the methodology section? I missed them.

(2) The authors state that they did not normalize various experiments to fungal mass because in all cases, the fungus had already covered the plate at 6 days. However, fungi can "cover a plate" and still have different growth profiles or mass. I agree with the authors that, in the case of their mutant, even if such differences affect their results, their observed effects will still likely be significant. However, the correct control would be (1) to grow the fungus for less time and see if there are differences before they "cover the plate"; and (2) Grow the fungi in liquid culture, and after 6 days- dry them and weigh them. I leave this to editorial decision as to whether it should be done.

Author Response

The authors have adequately answered most of my queries.

Two questions:

(1) Were the details of the RNAseq added to the methodology section? I missed them.

According to the reviewer’s comment, we have added the details of the RNA-Seq to the methodology section (lines 111-119). Small modifications were done in the Results section to avoid repeating information.

(2) The authors state that they did not normalize various experiments to fungal mass because in all cases, the fungus had already covered the plate at 6 days. However, fungi can "cover a plate" and still have different growth profiles or mass. I agree with the authors that, in the case of their mutant, even if such differences affect their results, their observed effects will still likely be significant. However, the correct control would be (1) to grow the fungus for less time and see if there are differences before they "cover the plate"; and (2) Grow the fungi in liquid culture, and after 6 days- dry them and weigh them. I leave this to editorial decision as to whether it should be done.

We agree with the reviewer in the need to normalize the data based on fungal biomass. We would like to point out that the results of the efeA gene expression during in vitro growth (Figure 2A) were normalized by using b-tubulin as a reference gene. The RNA-Seq data (Figure 2B) were also normalized. Gene expression data are presented as RPKM values, which are normalized based on the amount of reads mapped against the P. digitatum genome (Figure 2). However, the normalization of ethylene production is difficult according to our experimental design. In order to follow the production of ethylene along the experiment, we always have to use the same sample.  If we want to normalize the results according to the fungal biomass, we have to collect the biomass from the PDA plates, which implies that we cannot use these samples any longer. The suggestion of “growing the fungi in liquid media and, after 6 days dry them and weigh them” is a completely different experiment. If we do this, we also have to determine the ethylene production during growth in the liquid medium and, again, collecting the biomass implies the destruction of the sample.  In the previous manuscript revision, and according to this reviewer’s comments, we added Table 1 showing the mycelial growth of the parental strain, the ectopic mutant and two knockout mutants. There were no differences in growth among the different strains at neither  4 dpi nor 6 dpi, when ethylene production begins to be detectable in the parental strain and the ectopic mutant. Thus, we think that any possible difference in fungal biomass among the different strains would have a minor impact on "normalized" ethylene production.